# High-frequency PCR-testing as a powerful approach for SARS-CoV-2 surveillance in the field of critical infrastructure: A longitudinal, retrospective study in a German tertiary care hospital

**Bastian Fischer**[1]*, **Martin Farr**[1], **Jan Gummert**[2], **Cornelius Knabbe**[1], **Tanja Vollmer**[1]

**1** Herz- und Diabeteszentrum NRW, Institut für Laboratoriums- und Transfusionsmedizin, Bad Oeynhausen, Germany, **2** Herz- und Diabeteszentrum NRW, Klinik für Thorax- und Kardiovaskularchirurgie, Bad Oeynhausen, Germany

* bfischer@hdz-nrw.de

## Abstract

A high number of SARS-CoV-2 infections are mild, often even asymptomatic. Because of high specificity and sensitivity, RT-PCR is considered the gold-standard for COVID-19 testing. The technology played a key role in detecting sources of infection at an early stage and therefore preventing larger outbreaks. This was especially important in case of critical infrastructure, such as hospitals. Until now, comprehensive studies concerning the impact of high-frequency PCR-testing in German tertiary care hospitals during the COVID-19 pandemic are lacking. We therefore analyzed about 285.000 oral swab probes of 3.421 healthcare-workers concerning SARS-CoV-2 RNA positivity between November 2020 and February 2023. Our data show that frequent PCR-testing is a useful tool concerning SARS-CoV-2 surveillance. Due to the longitudinal character of the study, we were able to observe SARS-CoV-2 variant-specific differences. For example, the omicron-variant led to high reinfection-rates as well as lower Ct-values. Nevertheless, reinfection rates in our hospital are much lower compared to other analyzed healthcare-worker cohorts described in the literature, which is again attributable to the frequent testing-regime implemented in the early phase of the pandemic. Our data further reveal a longer infection-duration in elderly compared to younger individuals.

## Introduction

Severe acute respiratory syndrome coronavirus type-2 (SARS-CoV-2), causing coronavirus-disease-2019 (COVID-19), led to millions of deaths worldwide. In May 2023, the world health organization (WHO) declared that COVID-19 no longer constitutes a public health emergency of international concern. During the initial phase of the pandemic, the use of protective face-masks and strict non-pharmaceutical interventions (NPIs) seemed to be the most effective

**Data Availability Statement:** All relevant data are within the manuscript and its Supporting Information files.

**Funding:** The author(s) received no specific funding for this work.

**Competing interests:** The authors have declared that no competing interests exist.

tools to reduce viral-spreading. Over time, vaccines were developed showing partial effectiveness against transmission in addition to self-protection [1]. Due to genetic mutations, new viral-strains developed during the course of the pandemic, *inter alia* leading to vaccination-breakthroughs and higher transmission-rates [2]. In order to contain the spread of the virus, rapid antigen tests were offered in Germany at an early stage of the pandemic. Nonetheless, especially during periods of high SARS-CoV-2 prevalence, positive test-results have been frequently verified via PCR testing. The PCR-technique is the gold-standard in the diagnosis of COVID-19, as it is highly specific and sensitive [3]. In order to reduce viral transmission as well as socioeconomic losses, an optimized SARS-CoV-2 screening was crucial especially for system-relevant facilities, as for instance hospitals. Official and daily updated PCR-based data of the Robert-Koch-Institute (RKI), the public health institute for surveillance and prevention in Germany, were helpful to better assess the actual regional pandemic situation. Based on these data, institutions were for example able to adapt testing strategies.

The aim of this study was to monitor the monthly SARS-CoV-2 incidence within a tertiary care hospital between November 2020 and February 2023. Values were compared to those reported by the RKI, to better assess the number of unreported cases. Follow-up samples were analyzed to determine Ct-value progression as well as age- and sex-specific differences.

## Materials and methods

The heart and diabetes center (HDZ) is one of the worldwide leading tertiary care hospital specialized in diagnostics and therapy of cardiovascular diseases and diabetes. The HDZ employs around 2.500 people and is located in Bad Oeynhausen, a town in the district of Minden-Lübbecke, Germany. Between November 2020 and February 2023, we continuously monitored SARS-CoV-2 infection by analysis of about 285.000 oral swab samples of 3.421 HDZ health-care-workers with SARS-CoV-2 RNA positivity. The difference between individuals screened by PCR and employees is explainable by a regular staff turnover of employees per month accumulating over the 2.5 year period Samples for screening of individuals without suspicion of SARS-CoV-2 infection were screened in master pools of 10 individuals, which were set up by using a combination of 200 µl samples, respectively. Individuals of reactive pools were retested singularly. Individuals with suspicion of SARS-CoV-2 infection were directly screened individually. For the detection of SARS-CoV-2 RNA, the fully automated cobas (Roche, Switzerland) SARS-CoV-2 assay using the cobas 6800 automation system was used according to the manufacturers' instructions. The retrospective study was approved by the ethics committee of the medical faculty of the Ruhr University Bochum (AZ: 2023–1089). Data were initially accessed on 28.08.2023. Authors had no access to information that could identify individual participants during or after data collection. Therefore, the ethics committee waived the need for informed consent. GraphPad Prism 10.1.2 was used for the visualization and statistical analysis of the data.

## Results

Based on the official data of the RKI, our testing strategy was steadily adapted (Fig 1). While PCR-testing was offered on a voluntary basis in the initial phase of the pandemic, testing became mandatory from November 18, 2020 for 1st degree contacts. From the beginning of December 2020, anti-SARS-CoV-2 rapid antigen-tests were offered to employees. With the approval of the first coronavirus vaccines in Germany, employees were able to become vaccinated, starting in January 2021. A rise in COVID-19 case numbers in late summer 2021 led to a further adjustment of the testing regime. From September 20th, 2021 all unvaccinated employees had to conduct a PCR test twice a week. PCR testing became mandatory for all

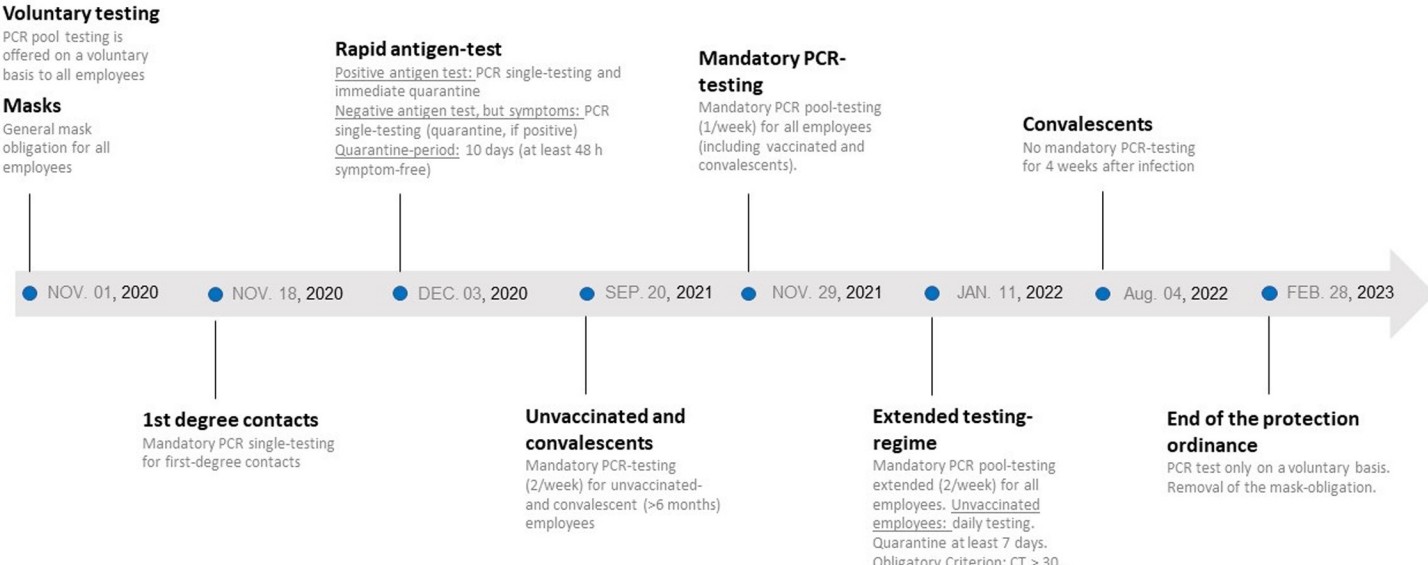

**Fig 1. Testing-regime of the HDZ-NRW during the COVID-19 pandemic in accordance with RKI suggestions.**

employees once a week starting at the end of November 2021. Testing regime was extended to two PCR tests per week for all employees from January 11, 2022. With the end of the protection-ordinance at the end of February 2023, mandatory testing and the requirement of wearing protective face-masks was waived.

As shown in Fig 2, testing frequency was highest from January 2022 until the end of the study, peaking in March 2022 (17.940 tests) and November 2022 (17.479 tests). The number of

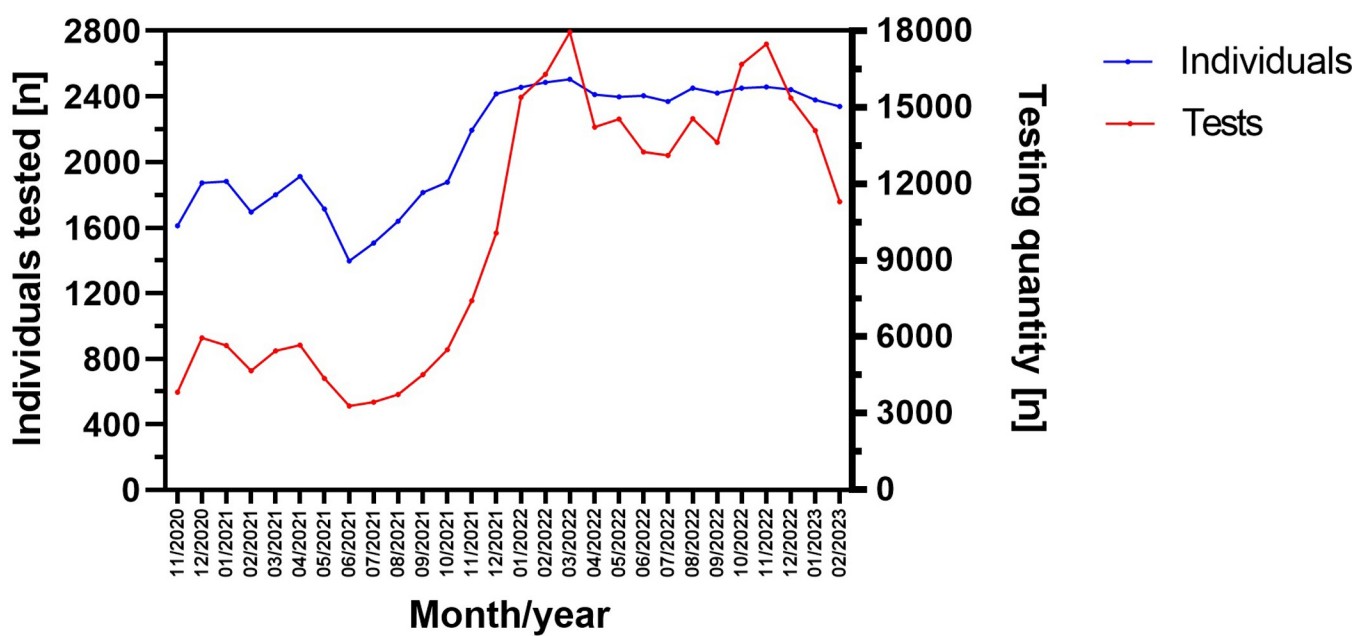

**Fig 2. Number of monthly PCR-tests and tested individuals during the study period between November 2020 and February 2023.**

healthcare-workers tested was dynamic during the study period, whereby values were highest between December 2021 (2415 individuals) and December 2022 (2442 individuals).

As shown in Fig 3, incidences were comparatively low and similar to those reported by the RKI until November 2021. Thereafter, the highly contagious Omicron-variant became dominant in Germany, which led to a strong increase in infection. In accordance with official data of the RKI (9.41%), our data reveal highest incidences in March 2022 (10.54%). During the following months, incidences remain on a high level, showing highest values in July (6.88%), October (7.83%) and December (7.21%) 2022. Starting in January 2022, our data consistently reveal higher incidences compared to those reported by the RKI (Fig 3).

Fig 4 shows monthly reinfection-incidendes normalized to both, total tested individuals as well as total employees. Due to the low rate of secondary- and third reinfections, the latter was only considered for the "1st reinfection cohort". First cases of reinfection were detected in December 2021 (reinfection-incidence: 5.88% and 0.04%, respectively). After a short decline, reinfection-rates steadily increased, peaking at the end of the study-period in February 2023 (reinfection-incidence: 45.71% and 2.05%, respectively). Our data also reveal secondary and third SARS-CoV-2 reinfections during the last months of the study (Fig 4).

We analyzed follow-up samples of 345 individuals (Fig 5, grey lines) to determine Ct-value progression after an initial positive SARS-CoV-2 test result. For the follow-up cohort, only individuals who submitted at least three further tests (within a maximum of 82 days) after an initial positive test were included. Furthermore, the initial Ct-value had to be < 30. The average course of this analysis is shown by the thick red line in Fig 5. Based on the regression line (red dotted line), we determined that the time between an initial positive test result and re-entry (Ct≥35) was on average 16 days.

Our data additionally show age-dependent differences in Ct-value progression after an initial positive SARS-CoV-2 test result. Thereby, the differences between the elevations of the

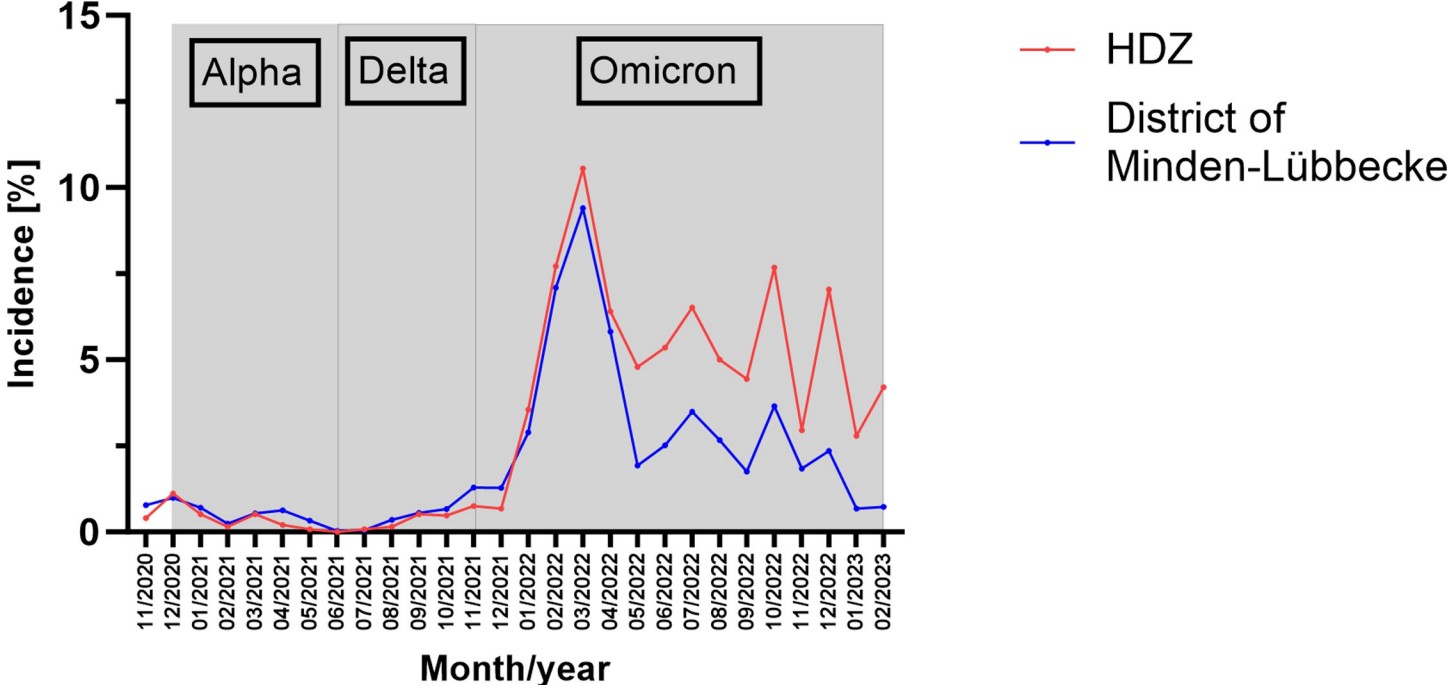

**Fig 3. Incidences based on our data (HDZ) compared to those of the RKI for the district of Minden-Lübbecke, Germany, between November 2020 and February 2023.**

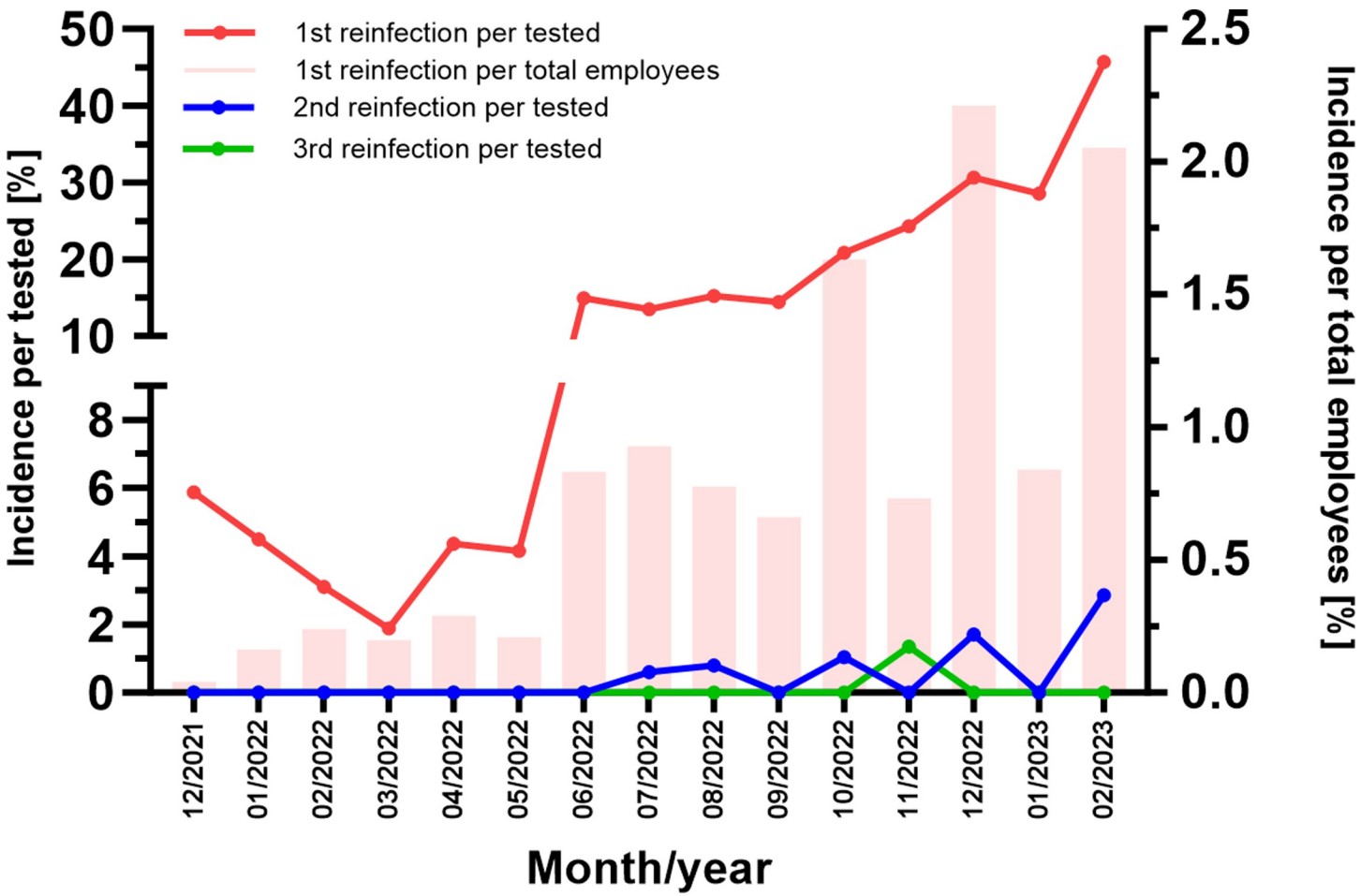

**Fig 4. Reinfection-incidences between December 2021 and February 2023 within our cohort.** Incidences were normalized to both, total tested individuals (left Y-axis, lines) as well as total employees (right Y-axis, bars).

age-groups 17–33 years and 49–70 years were highly significant (F = 10.48, DFn = 1, DFd = 19, P = 0.0043, Fig 6).

No sex-specific differences concerning Ct-value progression were observed (F = 0.6178, DFn = 1, DFd = 21, P = 0.4406, Fig 7).

## Discussion

On March 11, 2020, the world health organization (WHO) declared the COVID-19 outbreak a global pandemic. Since then, the causative SARS-CoV-2 virus caused millions of deaths and more than 750,000,000 infection-cases worldwide. Wearing face-masks was an early and efficient tool for containing the virus [4, 5]. In addition, non-pharmacologic interventions (NPIs) were implemented in many countries. Studies show, that the effectiveness of each NPI alone is limited and a combination of different NPIs is more purposeful [6]. Because of increased sensitivity and specificity, the use of PCR-based SARS-CoV-2 tests was an important factor in infection control. Another benefit is that the analysis can be carried out in high throughput [7]. Because of this, we offered voluntary PCR pool-testing to all employees already in November 2020. While incidences were comparatively low until December 2021, there was a sharp rise in cases from this point onwards. A first peak was reached in March 2022, with a positivity-rate

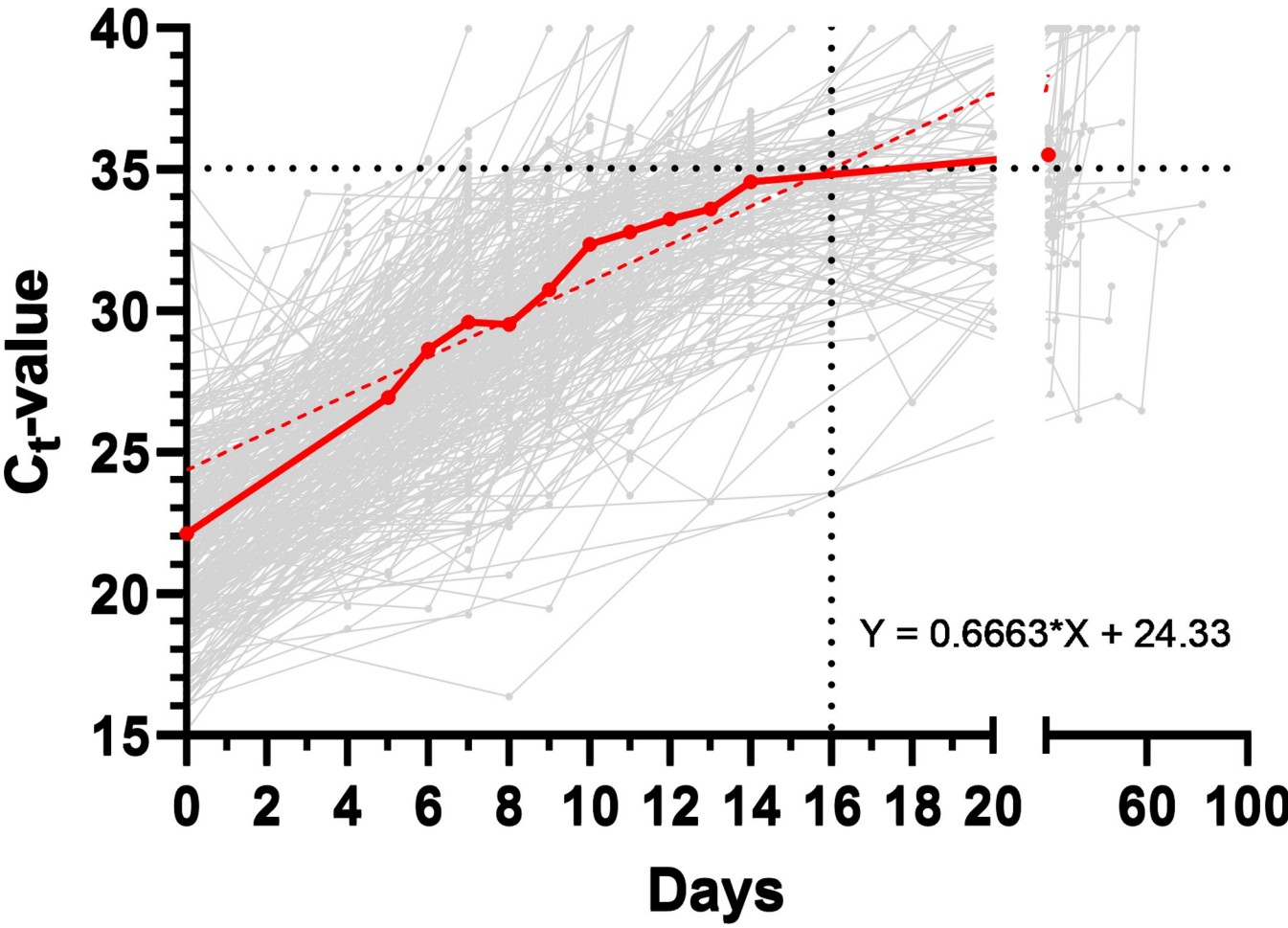

**Fig 5. Follow-up samples of 345 individuals to determine Ct-value progression after an initial positive SARS-CoV-2 test result.** The thick red line shows the average course of Ct-value progression (red dotted line: regression line.

of 10.54%. These observations are in line with those reported by the RKI, the German public health institute. The sharp rise in incidences is attributable to the Omicron-variant that predominated during this period. Transmission rates of the Omicron-variant were about 3.2 times higher than those of the previously dominant Delta-variant of the virus [8]. Weekly PCR testing was mandatory for all employees since end of November 2021. Our data show that the incidences determined from this point onwards were consistently higher than those of the RKI. This emphasizes the importance of high-frequency testing, which has been proven to lower the rate of undetected, asymptomatic cases and therefore has had a positive impact on outbreak control [9]. A study of Soni et al. additionally reveals that rapid antigen tests are not very useful concerning the detection of asymptomatic infections. In their study, 150 of 5,353 individuals were tested SARS-CoV-2 positive by PCR. However, only 10% of those who had no COVID-19 symptoms were detected by using a rapid test [10].

The first reinfection within our cohort occurred in December 2021 (reinfection-rate: 0.04%). From then on, reinfection rates increased steadily and peaked in December 2022 (2.2%). Previous studies already showed that reinfection rates increased, when the Omicron variant of the virus became dominant. A review of Nguyen et al., including 23,231 re-infected individuals, revealed a pooled reinfection-rate of 4.4%. Interestingly, the authors observed a

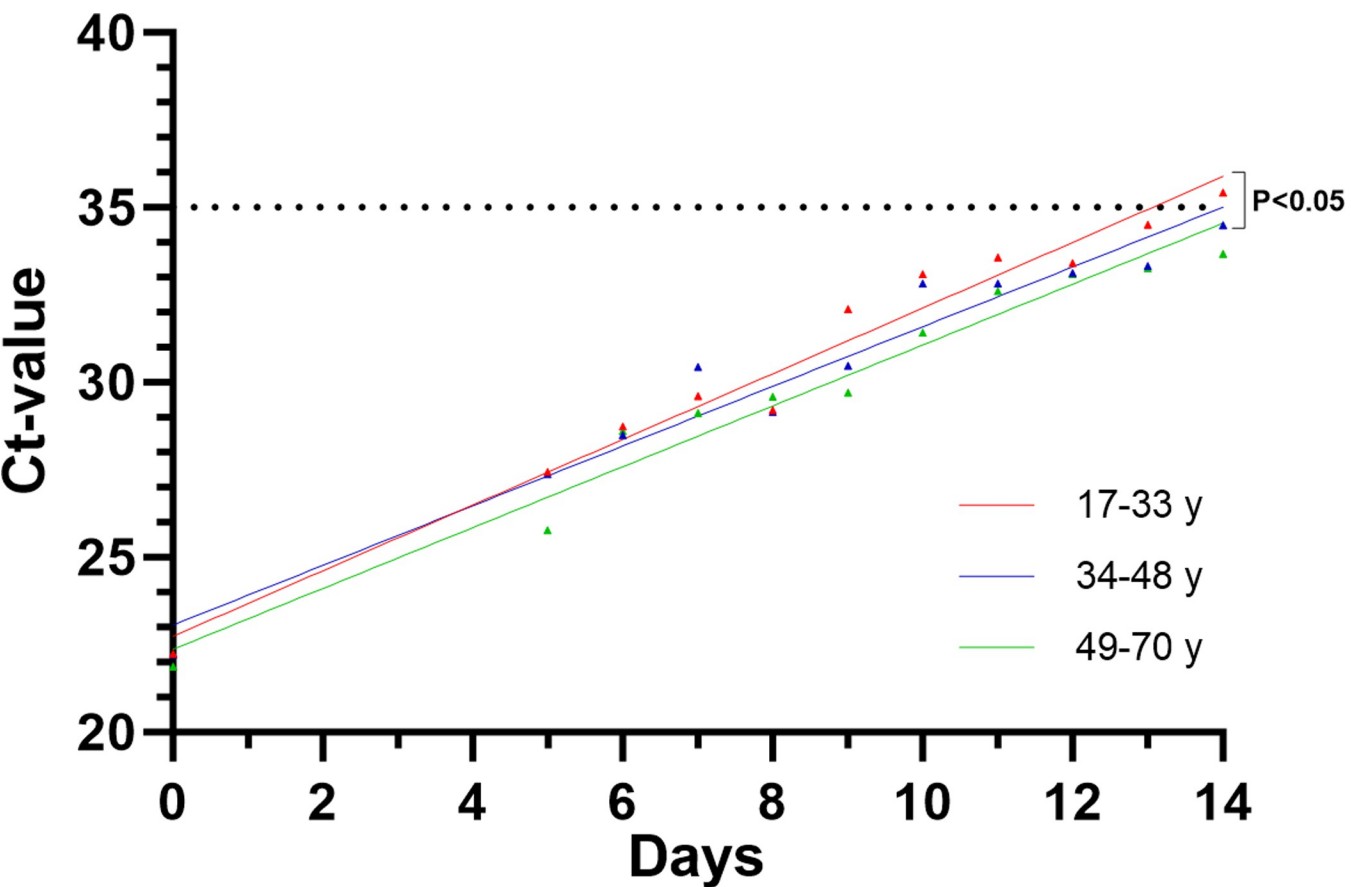

**Fig 6. Age-dependent Ct-value progression classified by three different age-groups.** Black dotted line represents the defined Ct-value negative-cutoff (Ct $\geq$35).

much higher prevalence in health-care-workers (6.8%) compared to the general population (2.9%) [11]. The comparable low reinfection-rates within our cohort are again attributable to the frequent test-regime. Another reason could be that wearing masks was made compulsory for employees at an early stage of the pandemic. Several studies show that masks reduce SARS-CoV-2 transmission [12–14].

Using follow-up samples, we estimated the Ct-value progression within our cohort. SARS-CoV-2 positive employees on average had an initial Ct-value of 22.1 and became negative (Ct $\geq$ 35) after 16 days. The comparatively low initial Ct-value is attributable to the Omicron variant, that the vast majority of employees were infected with. It was previously shown, that Omicron infections led to lower viral RNA peaks compared to Delta- and Alpha infections [15]. As for example, Waudby-West et al. calculated a median Ct-value of 28.7 within a Scottish community cohort between March and May, 2020, when the Alpha-variant of the virus dominated [16]. Hay et al. calculated a mean duration of 9.87 days for Omicron infections, using a negative Ct-cutoff value of $\geq$ 30. If this cutoff value is adopted to our dataset, our data are consistent with those of Hay et al. However, while other studies also used cutoffs between Ct 30 and 34 [17, 18], we chose the negative cutoff to be set at CT $\geq$ 35. A reason for this is that studies suggest intact viruses even in patients with higher Ct levels. Singanayagam et al. for example detected replicable viruses in 8% of samples showing a Ct-value > 35 [19].

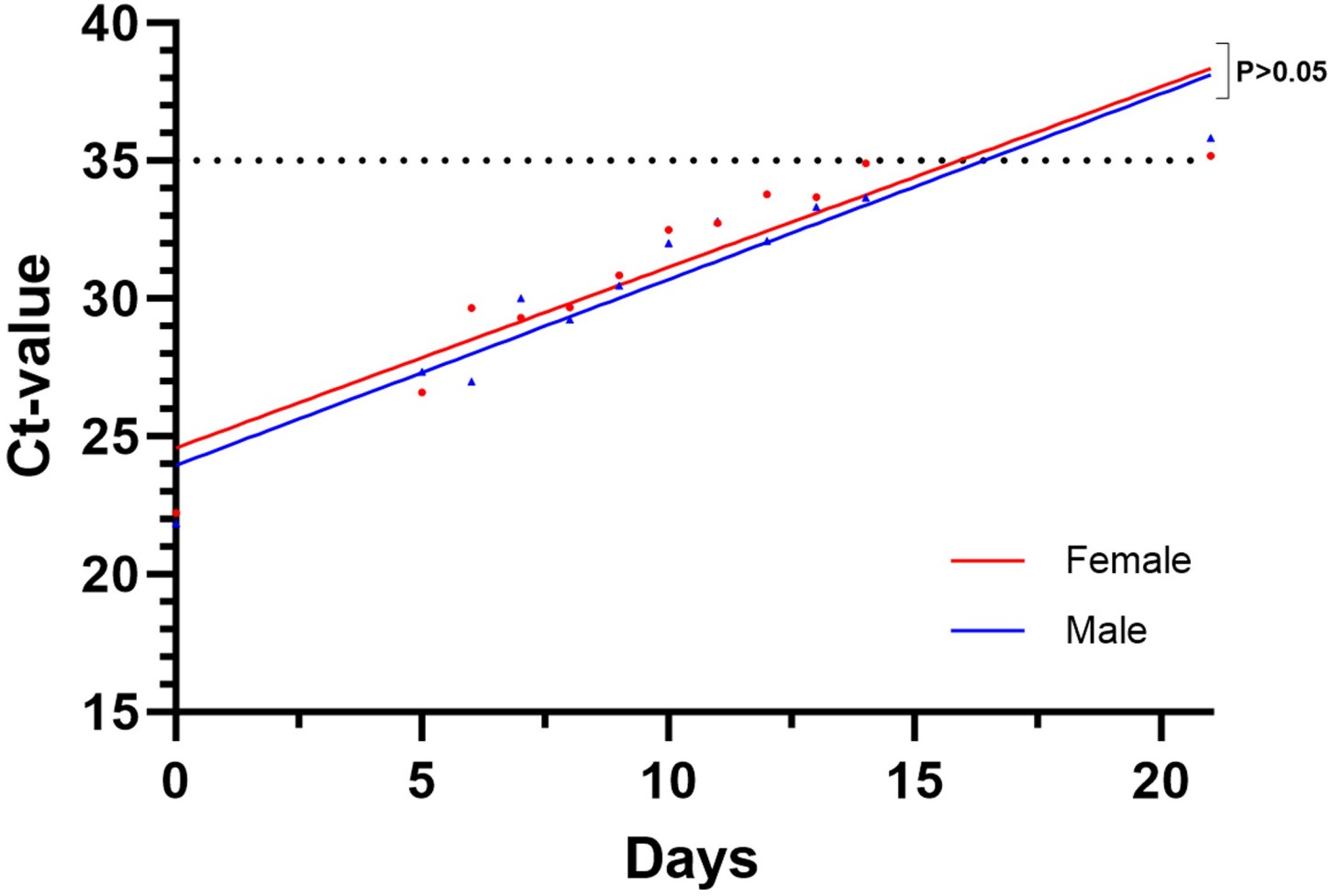

**Fig 7. Sex-dependent progression of Ct-values.** The black dotted line represents the defined Ct-value negative-cutoff (Ct $\geq$35).

While no sex-specific differences were observed, our data reveal significant age-dependent differences concerning Ct-value progression. The cohort aged between 49–70 years show a much higher progression compared to younger individuals (17–33 years). While little is known about this context in the literature, higher viral loads and a longer infection-duration in elderly were also seen before in a study of Mishra et al. [20].

Starting in January 2021, all employees had the opportunity to initially get vaccinated against SARS-CoV-2. A booster-dose was offered from November 2021 on. Our data do not allow a definitive conclusion about the impact of the vaccination campaign on the surveillance. However, a systematic review of Oordt-Speets et al. revealed a beneficial effect of vaccines on virus transmission. Vaccine effectiveness against transmission was ranging between 16–95%, primary depending on the viral-variant [21]. A recent study of Meslé et al. furthermore show that COVID-19 vaccines reduced deaths by 59% in the WHO European Region [22].

Before the first vaccines were available for the employees, NPIs were implemented with the aim of reducing viral transmission. Although the staff already follow a high standard of hygiene due to their profession, more extensive measures were imposed. These included, above all, an early, mandatory mask obligation and the avoidance of large gatherings. Earlier

studies already showed that wearing a face mask as well as physical distancing significantly lowers SARS-CoV-2 transmission [23, 24].

## Conclusion

High-frequency PCR-testing is a powerful tool concerning SARS-CoV-2 surveillance in the field of critical infrastructure. In addition, our data show that SARS-CoV-2 RNA is detectable for longer in elderly individuals compared to younger ones.

## Supporting information

**S1 Table. Number of monthly PCR-tests and tested individuals during the study period between November 2020 and February 2023 (raw values).**
(XLSX)

**S2 Table. Incidences based on our data (HDZ) compared to those of the RKI for the district of Minden-Lübbecke, Germany, between November 2020 and February 2023 (raw values).**
(XLSX)

**S3 Table. Reinfection-incidences between December 2021 and February 2023 within our cohort (Raw values).**
(XLSX)

**S4 Table. Follow-up samples of 345 individuals to determine Ct-value progression after an initial positive SARS-CoV-2 test result (raw values).**
(XLSX)

**S5 Table. Age-dependent Ct-value progression classified by three different age-groups (raw values).**
(XLSX)

**S6 Table. Sex-dependent progression of Ct-values (raw values).**
(XLSX)

## Author Contributions

**Conceptualization:** Bastian Fischer, Jan Gummert, Cornelius Knabbe, Tanja Vollmer.

**Data curation:** Tanja Vollmer.

**Formal analysis:** Bastian Fischer, Tanja Vollmer.

**Investigation:** Cornelius Knabbe.

**Methodology:** Bastian Fischer.

**Project administration:** Cornelius Knabbe.

**Supervision:** Jan Gummert, Cornelius Knabbe.

**Validation:** Martin Farr, Tanja Vollmer.

**Writing – original draft:** Bastian Fischer.

**Writing – review & editing:** Martin Farr, Jan Gummert, Cornelius Knabbe, Tanja Vollmer.

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
