## [Decision Letter · Decision Letter 0]

13 Aug 2024

PONE-D-24-10892High-frequency PCR-testing as a powerful approach for SARS-CoV-2 surveillance in the field of critical infrastructure: a longitudinal, retrospective study in a German tertiary care hospitalPLOS ONE

Dear Dr. Fischer,

Thank you for submitting your manuscript to PLOS ONE. After careful consideration, we feel that it has merit but does not fully meet PLOS ONE’s publication criteria as it currently stands. Therefore, we invite you to submit a revised version of the manuscript that addresses the points raised during the review process.

We look forward to receiving your revised manuscript.

Kind regards,

Giuseppe Di Martino

Academic Editor

PLOS ONE

Additional Editor Comments:

The paper is well written and clearly presented.

Major observations:

- In Methods section Authors should describe how data was presented and how analysis was conducted;

- Abput reinfections, Authors should report the incidence and in discussion section Authors should compare this result with published literature;

- Authors should also discuss how vaccination campaign impacted the surveillance.

- I suggest also to discuss the vaccination efficacy

Reviewers' comments:

Reviewer's Responses to Questions

**Comments to the Author**

1. Is the manuscript technically sound, and do the data support the conclusions?

Reviewer #1: Partly

Reviewer #2: Yes

2. Has the statistical analysis been performed appropriately and rigorously? 

Reviewer #1: Yes

Reviewer #2: Yes

3. Have the authors made all data underlying the findings in their manuscript fully available?

Reviewer #1: Yes

Reviewer #2: Yes

4. Is the manuscript presented in an intelligible fashion and written in standard English?

Reviewer #1: Yes

Reviewer #2: Yes

5. Review Comments to the Author

Reviewer #1: Overall, I believe that this is a well conducted study and a well written manuscript, which deserves to be published in the peer-reviewed evidence-based literature. As such, I have recommended that it be published with a couple of minor revisions.

I would greatly appreciate if the authors could please address the following concerns:

1. Most importantly, unless I am mistaken, I believe that your conclusion statement should read as "High-frequency PCR-testing is a powerful tool concerning SARS-CoV-2 surveillance in the field of critical infrastructure. In addition, our data show that SARS-CoV-2 RNA is detectable for longer in elderly individuals compared to younger ones" as opposed to "...for longer in younger individuals compared to elderly." After all, Figure 6 clearly shows that older individuals had lower CT values (higher viral loads) for a greater number of days than younger ones, and this phenomenon is also mentioned in your Discussion section.

2. Please fix the following minor grammatical error and clarify the following:

-"high-frequency PCR-testing" in place of "high-frequent PCR-testing" - in the Abstract section.

-"November 18, 2020" in place of "November 18th" for timing of when testing became mandatory - in the Results section.

Reviewer #2: The manuscript is meticulously crafted and well-organized, drawing from a sizable dataset of approximately 285,000 oral swab samples collected from 3,421 healthcare workers. It explores the impact of frequent PCR testing for SARS-CoV-2 surveillance during the COVID-19 pandemic. The statistical methods employed are robust and thoroughly documented, ensuring the reliability of the findings. The manuscript effectively communicates complex findings in an accessible manner, providing clear explanations for the study's choices and enhancing the overall clarity of the presentation.

The manuscript effectively utilizes a large dataset to address a critical issue in healthcare. While it is statistically robust and well-written, its retrospective design and setting-specific conclusions may limit its broader applicability. A more thorough discussion of potential biases, including the impact of external factors (e.g., changes in public health policies or vaccine uptake) on the results, would strengthen the manuscript.

The study cites 3,421 healthcare workers, while the Heart and Diabetes Center (HDZ) has around 2,500 employees. Could you clarify if the study included additional staff members beyond the healthcare workers, or if there might be another explanation for the difference in these figures?

6. PLOS authors have the option to publish the peer review history of their article (what does this mean?). If published, this will include your full peer review and any attached files.

Reviewer #1: No

Reviewer #2: **Yes: **Lawrence Annison

---

## [Author Response · Author response to Decision Letter 0]

15 Oct 2024

Response to reviewers

Additional Editor Comments:

The paper is well written and clearly presented.

Major observations:

- In Methods section Authors should describe how data was presented and how analysis was conducted;

Thanks, we added the information, that we used GraphPad Prism 10.1.2 for the visualization and statistical analysis of the data.

- About reinfections, Authors should report the incidence and in discussion section Authors should compare this result with published literature;

Thanks for this valuable comment. We reported the incidence within the “result” section (Fig. 4) and already compared our values with published literature. Based on your comment, we now calculated two different reinfection-incidences and supplemented Fig. 4 in this regard: the incidence per tested individual (Fig. 4, left Y-axis) and the incidence per total employees (Fig. 4, right Y-axis). Due to the low rate of secondary- and third reinfections, the latter was only considered for the “1st reinfection cohort”.

- Authors should also discuss how vaccination campaign impacted the surveillance.

We have now discussed this aspect in the last section of the discussion. 

- I suggest also to discuss the vaccination efficacy

We have now also discussed this aspect in the last section of the discussion.

Reviewers' comments:

Reviewer's Responses to Questions

5. Review Comments to the Author

Reviewer #1: Overall, I believe that this is a well conducted study and a well written manuscript, which deserves to be published in the peer-reviewed evidence-based literature. As such, I have recommended that it be published with a couple of minor revisions.

I would greatly appreciate if the authors could please address the following concerns:

1. Most importantly, unless I am mistaken, I believe that your conclusion statement should read as "High-frequency PCR-testing is a powerful tool concerning SARS-CoV-2 surveillance in the field of critical infrastructure. In addition, our data show that SARS-CoV-2 RNA is detectable for longer in elderly individuals compared to younger ones" as opposed to "...for longer in younger individuals compared to elderly." After all, Figure 6 clearly shows that older individuals had lower CT values (higher viral loads) for a greater number of days than younger ones, and this phenomenon is also mentioned in your Discussion section.

Thank you very much for this important suggestion. We have changed the sentence.

2. Please fix the following minor grammatical error and clarify the following:

-"high-frequency PCR-testing" in place of "high-frequent PCR-testing" - in the Abstract section.

-"November 18, 2020" in place of "November 18th" for timing of when testing became mandatory - in the Results section.

Thank you for your comments, we have made the changes.

Reviewer #2: The manuscript is meticulously crafted and well-organized, drawing from a sizable dataset of approximately 285,000 oral swab samples collected from 3,421 healthcare workers. It explores the impact of frequent PCR testing for SARS-CoV-2 surveillance during the COVID-19 pandemic. The statistical methods employed are robust and thoroughly documented, ensuring the reliability of the findings. The manuscript effectively communicates complex findings in an accessible manner, providing clear explanations for the study's choices and enhancing the overall clarity of the presentation.

The manuscript effectively utilizes a large dataset to address a critical issue in healthcare. While it is statistically robust and well-written, its retrospective design and setting-specific conclusions may limit its broader applicability. A more thorough discussion of potential biases, including the impact of external factors (e.g., changes in public health policies or vaccine uptake) on the results, would strengthen the manuscript.

Thanks a lot for your advice. We have expanded the discussion to include these aspects.

The study cites 3,421 healthcare workers, while the Heart and Diabetes Center (HDZ) has around 2,500 employees. Could you clarify if the study included additional staff members beyond the healthcare workers, or if there might be another explanation for the difference in these figures?

Thanks for this important demand. The general number of employees at our hospital is around 2,500. The difference between individuals screened by PCR and employees is explainable by a regular staff turnover of employees per month accumulating over the 2.5 year period. We added this aspect to the manuscript.

---

## [Decision Letter · Decision Letter 1]

20 Dec 2024

High-frequency PCR-testing as a powerful approach for SARS-CoV-2 surveillance in the field of critical infrastructure: a longitudinal, retrospective study in a German tertiary care hospital

PONE-D-24-10892R1

Dear Dr. Bastian Fischer,

We’re pleased to inform you that your manuscript has been judged scientifically suitable for publication and will be formally accepted for publication once it meets all outstanding technical requirements.

Kind regards,

Giuseppe Di Martino

Academic Editor

PLOS ONE

Additional Editor Comments (optional):

Reviewers' comments:

Reviewer's Responses to Questions

**Comments to the Author**

1. If the authors have adequately addressed your comments raised in a previous round of review and you feel that this manuscript is now acceptable for publication, you may indicate that here to bypass the “Comments to the Author” section, enter your conflict of interest statement in the “Confidential to Editor” section, and submit your "Accept" recommendation.

Reviewer #2: All comments have been addressed

2. Is the manuscript technically sound, and do the data support the conclusions?

Reviewer #2: (No Response)

3. Has the statistical analysis been performed appropriately and rigorously? 

Reviewer #2: (No Response)

4. Have the authors made all data underlying the findings in their manuscript fully available?

Reviewer #2: (No Response)

5. Is the manuscript presented in an intelligible fashion and written in standard English?

Reviewer #2: (No Response)

6. Review Comments to the Author

Reviewer #2: (No Response)

7. PLOS authors have the option to publish the peer review history of their article (what does this mean?). If published, this will include your full peer review and any attached files.

Reviewer #2: **Yes: **LAWRENCE ANNISON

---

## [Editor Report · Acceptance letter]

3 Jan 2025

PONE-D-24-10892R1 

PLOS ONE

Dear Dr. Fischer, 

I'm pleased to inform you that your manuscript has been deemed suitable for publication in PLOS ONE. Congratulations! Your manuscript is now being handed over to our production team.

Kind regards, 

on behalf of

Dr. Giuseppe Di Martino 

Academic Editor

PLOS ONE